# Peroxide Electrochemical Sensor and Biosensor Based on Nanocomposite of TiO_2_ Nanoparticle/Multi-Walled Carbon Nanotube Modified Glassy Carbon Electrode

**DOI:** 10.3390/nano10010064

**Published:** 2019-12-27

**Authors:** L. Andrés Guerrero, Lenys Fernández, Gema González, Marjorie Montero-Jiménez, Rafael Uribe, Antonio Díaz Barrios, Patricio J. Espinoza-Montero

**Affiliations:** 1Escuela de Ciencias Químicas, Pontificia Universidad Católica del Ecuador, Av. 12 de Octubre 1076, Apartado, Quito 17-01-2184, Ecuador; lenin.guerrero@yachaytech.edu.ec (L.A.G.);; 2School of Physics and Nanotechnology, Yachay Tech University, Urcuqui 100650, Ecuador; adiaz@yachaytech.edu.ec; 3Departamento de Química, Universidad Simón Bolívar, Caracas 89000, Venezuela; 4Instituto Venezolano de Investigaciones Científicas, Centro de Ingeniería Materiales y Nanotecnología, Caracas 1020-A, Venezuela; 5Departamento de Ingeniería Química, Escuela Politécnica Nacional, Quito 17-01-2759, Ecuador; rafael.uribe@epn.edu.ec

**Keywords:** electrochemical biosensor, hydrogen peroxide, titania-doped carbon nanotubes, cyclic voltammetry

## Abstract

A hydrogen peroxide (H_2_O_2_) sensor and biosensor based on modified multi-walled carbon nanotubes (CNTs) with titanium dioxide (TiO_2_) nanostructures was designed and evaluated. The construction of the sensor was performed using a glassy carbon (GC) modified electrode with a TiO_2_–CNT film and Prussian blue (PB) as an electrocalatyzer. The same sensor was also employed as the basis for H_2_O_2_ biosensor construction through further modification with horseradish peroxidase (HRP) immobilized at the TiO_2_–fCNT film. Functionalized CNTs (fCNTs) and modified TiO_2_–fCNTs were characterized by Transmission Electron Microscopy (TEM), Fourier Transform Infrared Spectroscopy (FTIR), and X-Ray DifFraction (XRD), confirming the presence of anatase over the fCNTs. Depending on the surface charge, a solvent which optimizes the CNT dispersion was selected: dimethyl formamide (DMF) for fCNTs and sodium dodecylsulfate (SDS) for TiO_2_–fCNTs. Calculated values for the electron transfer rate constant (ks) were 0.027 s^−1^ at the PB–fCNT/GC modified electrode and 4.7 × 10^−4^ s^−1^ at the PB–TiO_2_/fCNT/GC electrode, suggesting that, at the PB–TiO_2_/fCNT/GC modified electrode, the electronic transfer was improved. According to these results, the PB–fCNT/GC electrode exhibited better Detection Limit (LD) and Quantification Limit (LQ) than the PB–TiO_2_/fCNT/GC electrode for H_2_O_2_. However, the PB film was very unstable at the potentials used. Therefore, the PB–TiO_2_/fCNT/GC modified electrode was considered the best for H_2_O_2_ detection in terms of operability. Cyclic Voltammetry (CV) behaviors of the HRP–TiO_2_/fCNT/GC modified electrodes before and after the chronoamperometric test for H_2_O_2_, suggest the high stability of the enzymatic electrode. In comparison with other HRP/fCNT-based electrochemical biosensors previously described in the literature, the HRP–fCNTs/GC modified electrode did not show an electroanalytical response toward H_2_O_2_.

## 1. Introduction

The use of nanomaterials in biosensing is a subject of intense research, with a significant impact on everyday life [1]. Carbon nanotubes (CNTs) are at the forefront of this intense research, reflecting unique and remarkable mechanical, thermal, electrical, and elastic properties [2,3]. These properties include having characteristics of semiconductors or conductors [4]. CNTs have a significant potential application in nano-devices, particularly field-effect transistors [5,6] nano-probes [7], bioelectronics, sensors [8,9], and biosensors [10,11]. CNTs are used in biosensors to enhance bioanalytical performance or offer innovative routes of interfacing the transduction processes in the development of electrochemical biosensors [12]. It was reported that CNT-modified electrodes are widely used in the catalytic and detection electrochemistry of some interesting biomolecules [13,14,15,16]. Hydrogen peroxide is one of the most common analytes of biological interest because of its importance in biological, environmental, and industrial processes. In the human body, H_2_O_2_ can be converted into hydroxyl radicals (^•^OH), which are highly reactive [17]. The overproduction of reactive species such as OH and superoxide (O_2_)^⦁−^ in the cellular interstices is demonstrated to promote cell damage and tissue malfunction [18]. Also, H_2_O_2_ detection forms the diagnosis response of medical devices such as glucose sensors because, in the presence of oxygen, H_2_O_2_ is produced by the action of glucose oxidase [19,20]. Hydrogen peroxide is also found in food and drinking water, and it is also used in waste treatment and bleaching applications [21,22]. The presence of H_2_O_2_ in everyday life, such as in glucose monitors, food, bleaching processes, and even the regulation of human processes, highlights its important role in the detection of analytes. Therefore, the design of a biosensor should accomplish parameters such as versatility, high sensitivity, and fast response.

The successful application of CNT-based composite materials usually requires chemical modification of the CNTs [23,24]. Nanostructured metal oxides were found to exhibit strong adsorption ability, catalytic properties, and biocompatibility, making them ideal for sensor surface development and offering excellent interfaces for biological recognition [25]. Metal-oxide nanoparticles attached to the carbon nanotube surface are also receiving notable interest because of the potential applications when designing electrochemically functional nanostructures [2], which include a higher surface area and better biocompatibility, and they help in addressing the design of a biosensing interface allowing the analyte to interact effectively over the biosensing surface. Titanium dioxide received tremendous interest in promising areas such as photovoltaics, biosensing, and photocatalysis [26]. TiO_2_ nanostructures exhibit a large surface area, as well as unique chemical and electronic properties [27]. Considering electronic band structure, TiO_2_ is electron-rich, belonging to the n-type semiconductor category. TiO_2_ nanomaterials are used for sensors in the detection of soluble organics in aqueous media, as well as for gas, chemical, and biological substance detection [27].

One of the challenges in biosensing is to successfully design an interface between the analyte and the electrode surface [12]. In the matter of enzyme-based biosensors, the immobilization of the enzyme denotes an enormous task to lead the communication between its active site and the analyte. Furthermore, due to the remarkable properties mentioned above, the presence of modified CNTs could lead to an improved route of interfacing and a superior response in H_2_O_2_ detection. In this paper, the evaluation of TiO_2_ nanoparticles/multi-walled carbon nanotubes on glassy carbon (TiO_2_/MWCNT/GC) modified electrodes and Prussian blue (PB) helped us to design a model electrode to analyze the performance and contribution of carbon nanotubes and titanium dioxide nanostructures in the electrochemical detection of H_2_O_2_ by horseradish peroxidase (HRP)/TiO_2_/MWCNT/GC modified glassy carbon electrode.

## 2. Materials and Methods

### 2.1. Materials and Reagents

Nitric acid (HNO_3_, 68%) and hydrogen peroxide (H_2_O_2_, 30%) were purchased from Sigma-Aldrich Sigma, (Darmstadt, Germany). Potassium phosphate monobasic (KH_2_PO_4_) and sodium hydroxide (NaOH, 99.9%) were purchased from Fisher Scientific (Waltham, MA, USA). Phosphate-buffered saline (PBS, 20 mmol∙L^−1^ KH_2_PO_4_ + 20 mmol∙L^−1^ K_2_HPO_4_ + 0.1 mol∙L^−1^ KCl, pH 6.8) was used as a supporting electrolyte. Potassium ferricyanide (K_3_[Fe(CN)_6_]), iron trichloride (FeCl_3_), and potassium chloride (KCl) were from BDH Chemicals (Philadelphia, PA, USA), hydrochloric acid (HCl, 37%) was from Fisher Scientific (Waltham, MA, USA), carbon nanotubes were from Nanocyl (Sambreville Belgium), glassy carbon (GC), silver/silver chloride reference electrode, and graphite counter-electrode were from CH-Instruments (Austin, TX, USA), sulfuric acid (H_2_SO_4_, 98%) was from Fisher Scientific (Waltham, MA, USA), titanium isopropoxide, isopropanol (99%), and 1 µm, 0.3 µm, and 0.05 µm alumina powder were from CH-Instruments (Austin, TX, USA), dimethylformamide (DMF) was from BDH Chemicals (Philadelphia, PA, USA), and sodium dodecyl sulfate (SDS) and poly(diallyldimethylammonium chloride) (PDDA, 4% *w*/*w* in water) were from Sigma. Solutions (Darmstadt, Germany) were prepared using distilled/deionized water (18 MΩ resistivity).

### 2.2. Functionalization of Carbon Nanotubes

Pristine multi-walled carbon nanotubes (pCNTs) (NANOCYL^®^NC7000, Austin, TX, USA) with 90% purity, average diameter of 9.5 nm and average length of 1.5 μm, transition metal-oxide content <1%, surface area of 250 m^2^/g, and resistivity of cm 10^−4^ Ω∙m) were subjected to a pre-functionalization and functionalization process. To perform this work, the materials were obtained pre-functionalized and modified with TiO_2_ NPs. A brief description of the process is given below. Pre-functionalization was performed with 3 mol∙L^−1^ HNO_3_ and 1 mol∙L^−1^ H_2_SO_4_ solutions. Then, pCNTs were dispersed into the nitric acid solution while the sulfuric acid solution was slowly added. The solution was placed under reflux at 80 °C with stirring at 400 rpm for 6 h. Then, the nanotubes were filtered, rinsed, dried (12 h) and finally ground with a pestle and mortar. The pre-functionalization process caused the addition of carboxylic and hydroxyl groups to the nanotube surface. These groups were added to improve the compatibility with substrates and the media. The procedure was done by adding a 68% *w*/*w* HNO_3_ solution to pre-functionalized CNTs, submitted to sonication and stirring at 400 rpm at 80 °C for 2 h. The final suspension was filtered, rinsed, and dried (16 h). The resultant nanotubes were milled in a mortar and sieved with a 125 µm sieve.

### 2.3. Synthesis of TiO_2_–CNT Nanostructures

The synthesis was based on the sol–gel technique [28] using the previous fCNTs (150.0 mg), titanium isopropoxide as a precursor (0.6 mL), and isopropanol as a solvent (18.2 mL), with acetic acid and deionized water. The fCNTs were suspended into half the volume of isopropanol and sonicated for 30 min. A solution of titanium precursor and quarter the volume of isopropanol was prepared. The solution was dropped over half the volume of the fCNT suspension with constant stirring at 600 rpm. Then, a solution of the remaining isopropanol and deionized water was sonicated for 10 min and dropped into the main solution. The reaction was continued with the same parameters of stirring and temperature for 2 h. The suspension was left to age for 20 days at room temperature. The solvent was evaporated at 80–88 °C and washed three times with deionized water, letting it evaporate. The resultant precipitate was dried under vacuum at 80 °C for 4 h, followed by a thermal treatment at 500 °C in argon atmosphere for 2 h. The final substance was milled to obtain a thin powder.

### 2.4. Material Characterization

The TEM images were obtained in a JEOL 1220 microscope with an accelerating voltage of 100 kV. The samples were prepared using a wet suspension technique with an ethanol/water (70% *v*/*v*) solution. The FTIR spectra were obtained in a Nicolet iS10 FTIR spectrometer (Peabody, MA USA). The samples were prepared in KBr pills. The frequency range was from 4000 to 400 cm^−1^ with 64 sweeps at 2-cm^−1^ resolution. X-ray diffraction patterns were obtained in a SIEMENS D5005 diffractometer (Radeberg, Germany) with a 1.54178-Å wavelength in the range of 2θ = 10° to 80° with a rate of 0.02°/0.52 s. Zeta potential was determined for samples of fCNTs and TiO_2_–fCNTs. The measurements were done in an aqueous suspension of carbon nanotube samples with distilled water as the solvent. The environmental conditions of the analysis were a temperature of 24.1 °C and a humidity of 45.7%. Both samples were submitted to five measurement runs. From these data, we calculated the mean and standard deviation values from the zeta potential and half-width of the peaks.

### 2.5. Electrode Modification

The GC electrode surface is highly reactive, and impurities or different modifications could result in variations in its electrochemical activity [29]. These variations rely on the starting electrode surface. The most common electrode pretreatment to acquire reliable results is polishing. The polishing treatment removes the impurities and residues of the GC surface. The GC electrode of this work went through a polishing treatment each time it was modified. The treatment consisted of two processes: alumina polishing and electrochemical polishing. The GC electrode surface was polished with a polishing cloth in an aqueous slurry of 1 µm, 0.3 µm, and 0.05 µm of alumina powder performing eight-shaped motions for 5 min. After cleaning, the electrode was rinsed with distilled water, and then the GC electrode was assembled in a three-electrode cell containing 25 mL of 0.1 mol∙L^−1^ HNO_3_. The electrode was submitted to 50 cyclic voltammetry cycles from −1 V to 1 V at a scan rate of 100 mV∙s^−1^ with a negative initial polarization.

Functionalized CNTs were dispersed in DMF in a concentration of 5 mg∙mL^−1^, while TiO_2_–fCNTs were suspended in SDS at the same concentration as above. Both suspensions were placed in 1.5-mL tubes and sonicated for 10 min. Then, 10 μL of the fCNT and TiO_2_–fCNT dispersions were dropped onto the bare GC electrode to prepare the fCNT/GC and TiO_2_–fCNT/GC modified electrodes, respectively. The suspensions droplet must be uniformly dispersed over the electrode surface. The electrodes were dried at 50 °C for 15 min. Then, 10 µL of PDDA solution was dropped onto the fCNT/GC modified electrode surface and dried at 50 °C for 15 min. The suspension droplet was uniformly dispersed over the electrode surface. PDDA was not dropped onto the TiO_2_–fCNT/GC modified electrode surface. This was due to the different surface charge exhibited by the TiO_2_–fCNTs.

### 2.6. Electrodeposition and Activation of Prussian Blue at the Modified Electrode and HRP Immobilization at Modified Electrodes

A Prussian blue layer was deposited via the chronoamperometry technique onto the fCNT/GC and TiO_2_–fCNT/GC modified electrodes [30]. Electrodeposition was done by applying a constant potential of 0.4 V for 60 s, in a solution containing 2.5 mmol∙L^−1^ K_3_[Fe(CN)_6_] + 2.5 mmol∙L^−1^ FeCl_3_. After deposition, the PB film was activated in the 0.1 mol∙L^−1^ KCl + 0.1 mol∙L^−1^ HCl supporting electrolyte solution, which was used for film growth, by cycling the applied potential in a range of −0.2 to 0.8 V at a scan rate of 50 mV∙s^−1^.

A final layer of 10 µL of PDDA was dropped onto the modified electrodes surface and dried at 50 °C for 15 min [31,32]. After the last PDDA layer was placed and the whole modification was complete, the PDDA/PB/PDDA–fCNT/GC electrode was obtained. The final electrode arrangement was rinsed twice with distilled water. The whole modification process of the electrode is summarized in Scheme 1.

Finally, 10 μL of HRP (20 mg∙mL^−1^ in 0.1 mol∙mL^−1^ phosphate-buffered saline, pH 6.8) solution was dropped onto the modified electrode surface. The modified electrodes were stored at 4 °C for future use.

### 2.7. Electrochemical Characterization

Cyclic voltammetry (CV) and chronoamperometry experiments were performed with an electrochemical workstation CH-Instruments (Ch-instruments model 604 A Potentiostat). A conventional three-compartment electrochemical cell was employed with a modified GC electrode as the working electrode, graphite as the auxiliary electrode, and Ag/AgCl (saturated KCl) as the reference electrode from Bioanalytical Systems (BAS). The electrochemical behavior was investigated in PBS (pH 7). The CV was run over 20 cycles to reach stability of the electrode, i.e., where the current response did not change after a cycle.

### 2.8. Peroxide Detection

The electrocatalytic behavior of H_2_O_2_ at the modified electrodes was tested by chronoamperometry. Chronoamperometric measurements were carried out under stirred phosphate-buffered solution (0.1 mol∙L^−1^ phosphate-buffered solution, pH 7). In chronoamperometry, the current was monitored at a constant potential, while aliquots of H_2_O_2_ were injected into the pH 7.0 buffer every 40 s. Chronoamperometry was performed at different constant potentials, such as −0.2, −0.1, and 0 V vs. Ag/AgCl (saturated KCl).

## 3. Results and Discussion

### 3.1. Nanomaterial Characterization

#### 3.1.1. Transmission Electron Microscopy (TEM) 

Figure 1a exhibits the TEM images of functionalized CNTs. The shades on different sections of the image and the apparent change in gray tone indicate the bundling of the nanotubes over themselves. The mean diameter value of fCNTs was 5 ± 2 nm. In the literature, the value of wall separation distance of MWCNTs is around 0.34 nm [33]; thus, the nanotubes were around 4–10 layers thick. Figure 1b exhibits a multi-walled carbon nanotube, where approximately four walls can be seen with a diameter of 3 nm, supporting the previous calculation and the wall distance assumption. TiO_2_–fCNT nanostructure morphologies are displayed in Figure 2. In these images, titanium nanoparticles can be observed to be adhered onto the nanotube wall surface. The adhesion of the NPs is assumed to occur mainly via van der Waals interaction over the available functional groups from the previous functionalization; thus, NPs are not uniformly distributed over the nanotube surface. In some places, nanoparticles agglomerate over surfaces with higher numbers of functional groups and over the crossing of nanotubes, as shown in Figure 2b. The tendency to agglomeration is due to the high surface energy of titania nanoparticles; nevertheless, defined nanoparticles can still be seen. The particle sizes of the nanoparticles were measured from some TEM images, giving a mean value of 6.1 ± 1.3 nm. This value was close to the nanotube diameter.

#### 3.1.2. X-ray Diffraction (XRD)

Figure 3 shows the XRD analysis of TiO_2_–fCNTs (a) and fCNT (b) nanomaterials. The fCNT spectrum exhibited the characteristic 002 and 100 planes of graphite in nanotubes. These peaks were located at 2θ 26° and 43°, respectively. The 002 plane and its location indicate the interplanar distance of the nanotube layers. With *θ*_002_ = 26 and according to Braggs law, the interplanar distance was 3.43 Å. This calculation was done according to the previous assumption of the layer number calculation. The TiO_2_–fCNT diffractogram (Figure 3) shows the characteristic peaks at 26°, 38°, 48°, 54°, and 63° corresponding to the anatase crystalline phase [34]. Applying Scherrer’s equation [35], the average crystal size of the anatase nanoparticles was about 6.3 nm. This value was consistent with the particle size measured from the TEM images. Qualitatively, the anatase peaks were shown to be broad, which indicates the formation of crystals at the nanometric scale. The main peak of the 002 plane of fCNTs overlapped with the 101 plane of anatase in the TiO_2_–fCNT material.

#### 3.1.3. Fourier-Transform Infrared (FTIR) Spectroscopy

Figure 4 shows the FTIR spectra of (a) TiO_2_–fCNTs and (b) fCNTs. The TiO_2_–fCNT nanomaterial exhibited a band between 900 and 400 cm^−1^. This band refers to the Ti–O–Ti bond and proves the presence of titania, while this band was absent in the fCNT spectrum. Additional peaks were present along the spectra at 3433 cm^−1^, 3126 cm^−1^, 1388 cm^−1^, and 1061 cm^−1^. The fCNT spectrum exhibited a broad peak at 3280 cm^−1^, which is characteristic of the O–H stretching of a hydroxyl group and refers to the oscillation of carboxyl groups [36], equivalent to the 3433 and 3126 cm^−1^ peaks in the TiO_2_–fCNT spectrum. The peak at 2910 cm^−1^ refers to the methylene stretching band assumed to be groups located at defect sites in the CNT sidewalls [37], while the 1388 cm^−1^ peak could be attributed to rocking and bending of the –CH group due to possible hydrogenation in the functionalization process. C–O bands were observed in both spectra at 1061 cm^−1^, characteristic of carboxyl functional groups.

Functionalized CNTs and TiO_2_–fCNTs were characterized in a previous work by Albano [38] via N_2_ adsorption and Brunauer-Emmett-Teller (BET) area. In both cases, their results showed mesoporous structures with non-uniform pore sizes independent of fCNT quantity. In superficial area analysis, they showed an area of 298.40 ± 2.72 m^2^/g in the fCNTs, while, for TiO_2_–fCNTs, the area was 147.02 ± 1.89 m^2^/g. The presence of nanoparticles reduced the superficial area of the sample with respect to the fCNTs, due to the coverage of the TiO_2_ NPs on the CNT walls.

#### 3.1.4. Zeta Potential

Zeta potential characterization was done over nanotube samples, in order to analyze the charge over the electrical double layer. A potential difference was applied over the CNT suspensions to vary the mobility of the particles. This mobility provided information on the charge they present. The applied potential controlled the electrostatic interactions of the nanotubes and the solvent and, therefore, their colloidal stability [39]. Figure 5a shows the zeta potential curve of fCNTs in an aqueous solution. The mean value of the zeta potential was −51.84 mV with a standard deviation of 9.66 mV. Figure 5b shows the plot of the zeta potential measured from the TiO_2_–fCNT sample with a mean value of 9.93 mV and a standard deviation of 0.73 mV. The standard deviation values were calculated from five repeated measurements in the same conditions for both samples. In the functionalized CNTs, the previous acidic treatment over the pristine carbon nanotubes added carboxylic (–COOH) and hydroxyl groups (–OH) to their walls; thus, the negative charge was due to the deprotonation of the functional groups, and the fCNTs could be considered strongly anionic. According to Clogston and Patri [40], nanoparticles with values greater than ±30 mV are considered strongly cationic or anionic, while values within ±10 mV are considered neutral. In the TiO_2_–fCNT sample, the nanoparticles were attached to the functional groups as evidenced above, neutralizing the charge of the nanotubes and giving the sample a slightly positive charge. According to Patri et al. [40], TiO_2_–fCNTs can be considered neutral. Knowing the charge of the CNTs is essential for understanding the interaction between the particle and solvent and for predicting colloidal stability. The dispersion of the nanotubes plays a key role in the modification of the electrode surface. The selection of a solvent is quite a delicate issue depending on the charge of the nanotubes. Specific differences in CNT dispersion arise from the geometry and polarity of the solvent molecules or the solubility of the polymer [41]. As we can observe, if the charge of the samples differs, the solvent in which the carbon nanotubes are suspended is also different. DMF is a typical solvent for carbon nanotubes, and it was selected to suspend the fCNTs. The suspension presented stability after a few minutes of sonication due to the CNTs not spontaneously dispersing, as they required an input of energy to achieve dispersion. The interaction of the solvent molecules with the nanotubes involved weak polar forces and van der Waals interactions. The DMF molecule is planar and provides a better engagement through van der Waals interactions with the nanotube surface. The TiO_2_–fCNT sample presented a different charge than the fCNTs and required a different solvent. Vaisman et al. [42] explained the use of surfactants to disperse nanotubes depending on their charge and the type of surfactant. The selection of the surfactant depends on the charge that the CNTs exhibit. The TiO_2_–fCNT sample was dispersed in a solution of SDS. SDS is an anionic surfactant with a sulfate group at its head, which gives it the amphiphilic properties. SDS debundles nanotubes through steric and electrostatic interactions. The anionic head of the surfactant molecule tends to attach to the nanotube walls and to be adsorbed over the forming micelles, providing colloidal stabilization.

#### 3.1.5. Cyclic Voltammetry (CV)

Figure 6 shows the cyclic voltammograms (CVs) at the GC electrode, fCNT/GC, and TiO_2_–fCNT/GC modified electrodes in 0.1 mol∙L^−1^ phosphate-buffered solution at pH 7. At the GC electrode, Figure 6a does not exhibit any oxidation/reduction reaction in the electrolytic medium and very low non-faradaic current is shown in this potential region, exhibiting an almost flat current response. The addition of fCNTs to the GC electrode (Figure 6b) showed a strong increment of the non-faradic current with respect to the bare electrode. The non-faradaic current does not involve any electron transfer; it only causes the accumulation of electric charges as potential is applied [43]. This result shows that functionalized CNTs exhibit a highly capacitive ability. On the other hand, the presence of peaks at 0–0.3 V meant that there was at least a reaction associated with the oxidation and reduction of the –OH and –COOH functional groups, leading to the formation of quinones around the surface of the nanotubes. When the GC electrode was modified with TiO_2_–fCNTs (Figure 6c), the signal was similar with respect to the bare GE. There were no signals associated with electroactive species, and the non-faradaic current was a little higher with respect to the fCNT/GC modified electrode, suggesting that modified nanotubes do not exhibit capacitive properties as sturdy as functionalized ones. The lack of the pair of peaks around 0 and 0.3 V presented in the fCNT/GC modified electrode confirmed the results obtained by the zeta potential studies, whereby a chemical interaction was present between the TiO_2_ nanostructures and the –OH and –COOH functional groups previously generated with the acidic treatment over the pristine carbon nanotubes.

The electrochemical behavior of the modified electrodes, i.e., PB–fCNT/GC and PB–TiO_2_/fCNT/GC, was monitored by cyclic voltammetry experiments (Figure 7). At both electrodes, two pairs of redox waves appeared, and the cyclic voltammetry results were in agreement with the literature [44,45]. The pair of peaks between +0.09 and +0.25 V (when reduced to Prussian white) resulted from the redox reactions of low-spin Fe(CN)_6_^3−/4−^, and another pair of peaks, appearing between +0.77 and +0.92 V (when oxidized to Prussian yellow) corresponded to electrochemical reactions of high-spin Fe^3+/2+^ [44,46]. In a reversible reaction, ∆E must be constant and independent of the scan rate, with a value close to 0.059 V, and the I_pa_/I_pc_ peack ratio must be equal to 1. In this way, at the PB–fCNT/GE modified electrode, in the reduction zone, the peak ratio was 0.58, which indicates non-complete reversibility at the electrode, while ∆E was +0.21 V, showing a higher resistance produced by the layers; in the oxidation zone, the peak ratio was +0.38, and ∆E was +0.13 V. The PB–TiO_2_/fCNT/GC modified electrode, in the reduction zone, displayed an I_pa_/I_pc_ ratio of 0.8, and the ∆E was +0.35 V; in the oxidation zone, the I_pa_/I_pc_ ratio was +1.4, and the ∆E was 0.1 V. The values obtained on the modified electrodes predicted the quasi-reversible reaction processes of the PB at the electrode surfaces. For both modified electrodes, the redox peak potentials were dependent on scan rate (not shown). It was observed that the values of E_pa_ and E_pc_ shifted slightly in the positive and negative directions, respectively, whereas the ΔEp increased with the increase in scan rate, but E_1/2_ was almost independent of the scan rate. The anodic and cathodic peak currents were linearly proportional to the scan rate up to 350 mV∙s^−1^ (not shown), suggesting that the electrochemical behavior of PB for both composite films (fCNT and TiO_2_/fCNT) was not a diffusion-controlled process but a typical surface-controlled one. Based on the Laviron theory [47,48], the electron transfer rate constant (ks) was determined for modified electrodes in the phosphate-buffered solution by measuring the variations of the peak potentials of the anodic and cathodic peaks at different scan rates, when ΔEp was larger than 100/*n* mV. Assuming *n* = 4, the calculated values for ks were 0.027 s^−1^ at the PB–fCNT/GC modified electrode and 4.7 × 10^−4^ s^−1^ at the PB–TiO_2_/fCNT/GC electrode, suggesting that, on the PB–TiO_2_/fCNT/GC modified electrode, the electronic transfer was improved. The surface concentration (***Гc***) of electroactive species at the modified electrodes was evaluated from the slope of ***Ip***/***A*** versus ν, in which ***A*** is the surface area, according to following equation [49]:(1)$$Ip=\frac{n^{2}F^{2}A\nu \;Гc}{4RT}.$$

The average value of ***Гc*** for the redox peaks was 4.98 × 10^−9^ mol∙cm^−2^ and 4.72 × 10^−9^ at the PB–fCNT/GC and PB–TiO_2_/fCNT/GC modified electrodes, respectively (for ***n*** = 4 and ***v*** < 1000 mV∙s^−1^). The similarity in the values obtained suggests that TiO_2_ was responsible for improving the electronic transfer on the electrode.

### 3.2. Electrochemical Performances of the PB–fCNT/GC and PB–TiO_2_/fCNT/GC Modified Electrodes toward Hydrogen Peroxide

Figure 8 shows the cyclic voltammograms of H_2_O_2_ at the PB–TiO_2_/fCNT/GC electrode, in the zone of PB redox reactions of low-spin Fe(CN)_6_^3−/4−^, Figure 8a, and the zone of PB redox reactions of high-spin Fe^3+/2+^, Figure 8b. The literature reports [44] that, at the bare electrode, H_2_O_2_ starts to oxidize only above +0.9 V and starts to reduce below −0.1 V, while no redox peaks can be seen within the applied potential. In contrast, at the PB–TiO_2_/fCNT/GC modified electrode, a cathodic response and an anodic response to H_2_O_2_ started to occur at +0.2 V and +0.7 V, respectively. Both zones of the PB film voltammogram showed a marked decrease in the reverse redox currents and a large increase in the forward redox currents, demonstrating that the electrocatalytic reduction of hydrogen H_2_O_2_ was effective in both zones. The corresponding cyclic voltammograms indicated that two pairs of redox peaks of the PB film catalyzed independently of the oxidation and reduction of H_2_O_2_. The redox peaks at +0.93 V exhibited an electrocatalysis toward the oxidation of H_2_O_2_, while the redox peaks at +0.1 V catalyzed the reduction of H_2_O_2_. Cyclic voltammograms of H_2_O_2_ at the PB–fCNT/GC electrode (not shown) also showed that this modified electrode could catalyze both oxidation and reduction.

The chronoamperometric detection of H_2_O_2_ at the PB–TiO_2_/fCNT/GC electrode was performed at constant potentials of 0 V (Figure 9a) and +1 V (Figure 9b). The current response was linear for H_2_O_2_ concentrations in the range of 0.5 to 4.3 mmol∙L^−1^ (*y* = −10.428*x* − 94.82, *R*^2^ = 0.997) and 0.04 to 0.9 mmol∙L^−1^ (*y* = 0.0002*x* + 3 × 10^−5^, *R*^2^ = 0.991). The quantification limit (LQ) was 0.29 mmol∙L^−1^, and the detection limit (LD) was 0.088 mmol∙L^−1^ in the oxidation zone, while, in the reduction zone, the LQ was 0.31 mmol∙L^−1^, and the LD was 0.092 mmol∙L^−1^. According to these results, the PB–TiO_2_/fCNT/GC modified electrode can be used to detect peroxide both in the reduction zone and in the oxidation zone. The chronoamperometric detection of H_2_O_2_ at the PB-fCNT/GC, at constant potentials of 0.0 V and +1 V, is shown in Figure 9c,d, respectively. The current response was linear for H_2_O_2_ concentrations in the range of 0.05 to 0.8 mmol∙L^−1^ (*y* = −163.01*x* − 8.5347, *R*^2^ = 0.992) and 0.05 to 0.9 mmol∙L^−1^ (*y* = 1.66*x* − 1.05, *R*^2^ = 0.997). An LQ of 0.079 mmol∙L^−1^ and an LD of 0.024 mmol∙L^−1^ were obtained in the oxidation zone, while the LQ was 0.051 mmol∙L^−1^, and the LD was 0.015 mmol∙L^−1^ in the reduction zone. According to these results, the PB–fCNT/GC electrode showed a better LD and LQ than the PB–TiO_2_/fCNT/GC electrode. However, at the PB–fCNT/GC electrode, the potential affected the stability of the PB film. The PB film was very unstable at the potentials used. Therefore, the PB–TiO_2_/fCNT/GC modified electrode was considered best for H_2_O_2_ detection in terms of operability.

To investigate the reproducibility of the modified electrodes, 10 electrodes made with independently the same electrode showed acceptable reproducibility with a relative standard deviation average of 3.6% for the current determined at 2 mol∙L^−1^ H_2_O_2_. When the modified electrodes were not in use, they were stored in PBS (pH 7.0) at 4 °C. To examine the long-term storage stabilities, the voltammetric responses of the modified electrodes to 2 mol∙L^−1^ H_2_O_2_ were monitored with respect to the storage time every five days. After a 30-day storage period, the PB–TiO_2_/fCNT/GC electrode still retained 80% of its initial current response to H_2_O_2_ which indicated that the enzyme electrode had good stability, while the PB–fCNT/GC electrode only retained 15% of its initial current response to the analyte.

### 3.3. Electrochemical Performance of TiO_2_/fCNT/GC Modified Electrode as an Enzyme-Based Electrochemical Biosensor

The TiO_2_/fCNT/GC modified electrode was also employed as the basis for designing H_2_O_2_ biosensors through further modification with HRP immobilized at the TiO_2_/fCNT film. The chronoamperometric responses obtained at the HRP–TiO_2_/fCNT/GC electrode following successive additions of H_2_O_2_ are shown in Figure 10a. The HRP biosensor exhibited a very fast response to H_2_O_2_, reaching about 90% of the steady-state signal within 10 s. This biosensor was able to detect H_2_O_2_, exhibiting a linear range of response (*r* = 0.9997, *n* = 5) between 0.5 mmol∙L^−1^ and 7.5 mmol∙L^−1^, according to the following equation: *y* = −9.63*x* + 17.06 (Figure 10b). The biosensor showed a limit of quantification of 2.69 mmol∙L^−1^ and a detection limit of 0.81 mmol∙L^−1^ at a signal-to-noise ratio of 3. The electrocatalytic performance of the HRP–TiO_2_/fCNT/GC electrode toward the reduction of H_2_O_2_ was investigated by cyclic voltammetry. Figure 10c shows the CV behaviors of the HRP–TiO_2_/fCNT/GC modified electrode before (a) and after (b) the chronoamperometric test (50 cycles), where it can be observed that the voltammetric currents did not change, suggesting the high stability of the enzymatic electrode. In comparison with other HRP–fCNT-based electrochemical biosensors previously described in the literature, it should be noted that the HRP–fCNT/GC modified electrode presented in this paper did not show an electroanalytical response toward H_2_O_2_. 

### 3.4. Comparison of Results with Previous Sensors Published in the Literature

Table 1 shows a comparison between PB–fCNT/GC and HRP–TiO_2_/fCNT/GC electrodes and other electrodes reported in in the literature. It can be seen that our results are in line with other reports, suggesting that this electrode coating can be used to sense H_2_O_2_.

## 4. Conclusions

An electrochemical sensor, PB–TiO_2_/fCNT/GC, and an electrochemical biosensor, HRP–TiO_2_/fCNT/GC, for H_2_O_2_ detection were efficiently designed with MWNT modified GC electrodes. Calculated values for ks were greater at the PB–fCNT/GC electrode than at the PB–TiO_2_/fCNT/electrode, suggesting that TiO_2_ at the fCNT/GC modified electrode improved the electronic transfer due to the characteristics of TiO_2_. In addition, Γ values were equal for both modified electrodes (PB–fCNT/GC and TiO_2_/fCNT/GC). The PB–TiO_2_/fCNT/GC modified electrode was considered the best for H_2_O_2_ detection in terms of operability, while HRP–TiO_2_/fCNT/GC modified GC electrodes showed an electroanalytical response toward H_2_O_2_.The results demonstrated that the nanostructured environment of the TiO_2_-based film was a suitable matrix for the immobilization of HRP to retain its activity. The long-term stability of the HRP–TiO_2_/fCNT/GC modified electrode was attributed to the excellent biocompatibility of the nanostructured TiO_2_.

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
