# Peer review of "Peroxide Electrochemical Sensor and Biosensor Based on Nanocomposite of TiO2 Nanoparticle/Multi-Walled Carbon Nanotube Modified Glassy Carbon Electrode"

_nanomaterials, 2019, doi:10.3390/nano10010064_

Round 1
Reviewer 1 Report
Very good paper, well written, with original results, deserving publication. Some typos should be corrected throughout the text; I recommend a careful re-reading of the text by the Authors. Here I mention only a few of them. In the Abstract:
did not showed-> did not show
Please define abbreviations in the Abstract: PB, LD, LQ
Line 205: did not bond -> did not bind
In the Conclusions:
modified electrodes was -> modified electrodes were
Author Response
Author's Reply to the Review Report (Reviewer 1)
Very good paper, well written, with original results, deserving publication. Some typos should be corrected throughout the text; I recommend a careful re-reading of the text by the Authors. Here I mention only a few of them. In the Abstract:
did not showed-> did not show
The observations were included in the final document
Please define abbreviations in the Abstract: PB, LD, LQ
The observations were included in the final document
Line 205: did not bond -> did not bind
The observations were included in the final document
In the Conclusions:
The observations were included in the final document
Modified electrodes was -> modified electrodes were
The observations were included in the final document

Reviewer 2 Report
There are several places where sloppy editing is in evidence. Abbreviations need a careful revision, e.g., in Introduction (line 88), “…carbon nanotubes and titanium dioxide…” were previously described and abbreviated but the authors did not follow.
The authors described the electrodeposition of Prussian blue in the experimental sub section (line 168) “…applying two consecutive pulses at constant potential of 0.4 V by 60 seconds…” this is not clear, how is it applied two pulses?
In Line 176, the authors described “The whole modification process of the electrode is summarized in the Scheme 1” however, the scheme is not included in the manuscript.
Figure 8 is not explained clearly: what is the difference between three color lines in the figure 8a and 8b?
How is the enzyme, HRP, immobilized on the electrode?
The authors should insert a table comparing their results with previously published one using similar electrode modifications in order to clearly show what is the advantage of their modification and results.
In sum, additional studies such as selectivity and reproducibility should be conducted or clearly discussed to prove the enhanced electrocatalytic activity occurred in their system with experimental results.
Author Response
Author's Reply to the Review Report (Reviewer 2)
There are several places where sloppy editing is in evidence. Abbreviations need a careful revision, e.g., in Introduction (line 88), “…carbon nanotubes and titanium dioxide…” were previously described and abbreviated but the authors did not follow.
Abbreviations are followed in the new version of the manuscript
The authors described the electrodeposition of Prussian blue in the experimental sub section (line 168) “…applying two consecutive pulses at constant potential of 0.4 V by 60 seconds…” this is not clear, how is it applied two pulses?
The electrodeposition of the PB on the modified electrode surfaces has been clarified in section 2.6 of the new version of the manuscript:
"Electrodeposition was done by applying a constant potential of 0.4 V within 60 s, in a solution contained 2.5 mmol L-1 K3[Fe(CN)6] + 2.5 mmol L-1 FeCl3. After deposition the PB film was activated in 0.1 mol L-1 KCl + 0.1 mol L-1 HCl supporting electrolyte solution, which was used for film growth, by cycling the applied potential in a range of -0.2 to 0.8 V at a scan rate of 50 mV s-1⦋32⦌."
In Line 176, the authors described “The whole modification process of the electrode is summarized in the Scheme 1” however, the scheme is not included in the manuscript.
Scheme 1 has been includes in section 2.6 of the new version of the manuscript. Authors apologize for the omission we have committed
Figure 8 is not explained clearly: what is the difference between three color lines in the figure 8a and 8b?
Figure 8 legend has been improved in the new version of the manuscript: Figure 8. Cyclic voltammograms of H2O2 at PB-TiO2/fCNT/GC electrode, in the PB redox reactions of low spin Fe(CN)6 3-/4- (a) and the PB redox reactions of high spin Fe3+/2+ (b). Scan rate 50 mV s-1, PB buffer. Without H2O2 (black line), 1 mol L-1 H2O2 (red line) and 2 mol L-1 H2O2 (blue line)
How is the enzyme, HRP, immobilized on the electrode?
Explanation of how HRP was immobilized on the electrode surface have added in the new version of the manuscript (Section 2.7).
The authors should insert a table comparing their results with previously published one using similar electrode modifications in order to clearly show what is the advantage of their modification and results.
Comparison of the characteristics of the modified electrodes with previous reports has been added in the new version of the manuscript (Section 3.4).
In sum, additional studies such as selectivity and reproducibility should be conducted or clearly discussed to prove the enhanced electrocatalytic activity occurred in their system with experimental results.
Studies about reproducibility for HRP-TiO2/fCNTs/GC before (a) and after (b) of the test chronoamperometric were reported in Fig.10 c. First, the modified electrode was evaluated by examining the cyclic voltammetric peak currents after continuous scanning for 50 cycles. There was nearly no decrease in the voltammetric response, indicating that HRP-TiO2/fCNTs/GC electrode was stable in buffer solution. While studies about reproducibility for PB-TiO2/fCNTs/GC and PB/fCNTs/GC modified electrodes are now discussed in the new version of the manuscript (Section 3.2).
Selectivity studies we consider irrelevant, since we are using a purely selective peroxide system with the use of HRP and/or PB.

Reviewer 3 Report
The manuscript describes a hydrogen peroxide (H2O2) sensor and biosensor based on modified multi-walled carbon nanotubes (CNTs) functionalizated with titanium dioxide (TiO2) nanostructures. Glassy carbon electrode was modified with TiO2-CNTs film and prussian blue as electrocatalyzer. The same sensor was also employed as the basis for H2O2 biosensors construction through further modification with Horseradish Peroxidase (HRP) immobilized at TiO2-CNTs film. Functionalized CNTs and modified TiO2-fCNTs were characterized by TEM, FTIR and XRD. Zeta potential was used to determine the surface charge on both CNTs samples. Solvent was optimized depending on the surface charge. PB-TiO2/fCNTs/GC modified was considered the best for the H2O2 detection in terms of operability. HRP-TiO2/fCNTs/GC modified GC electrodes showed electroanalytical response toward H2O2.
The manuscript is well organized, the introduction contextualizes the experimental section, the results section is concise, but the conclusion section could be extended by highlighting the advantages of this sensor and biosensor compared to existing (bio)sensors.
I recommend this paper for publication in Nanomaterials. Nonetheless, some major changes must be covered:
On abstract, I think it would be appropriate to summarize the abstract.
On introduction section: The authors should extend the introduction including information and some references about composite electrodes based on graphite modified with glucose oxidase to glucose determination. This is a simple alternative to biosensors development. Some references are: Biosensors and Bioelectronics 78(2016)505–512; Microchemical Journal 119 (2015) 66-74.
On experimental section: Authors should complete the description of the materials and methods section. For example: purity of reagents, Sigma carbon nanotubes physical characteristics (as diameter and long) are necessary but they are not showed.
On line 109 (page 3). I think that the correct description for “pC-NTs dissolved” is “pC-NTs dispersed”. Please, specify the Nanocyl carbon nanotubes physical characteristics too.
On lines 178-183 (page 4). Authors should complete the description (model) of the equipment used. Information about reference electrode should be extended (model and internal solution concentration). The same for lines 190 of page 5.
On results section: The morphological and physical characterization of the different nanomaterials is complete and rigorous. Some figures could be removed anyway. For example, Figures 3, 4 and 5 could be placed in the supplementary section. The caption of Figures 1 and 2 should be completed. Images a and b are not detailed.
On line 368-369 (page 12) the authors said that “The chronoamperometric detection H2O2 at PB-TiO2/fCNTs/GC electrode was performed at constant potentials of 0 V (Figure 9a) and +1 V (Figure 9b)”. Applied potential on oxidation process is very high and other species could be oxidized. Could the authors extended this part? Was an interference study conducted? How was the detection limit determinated?
On section 3.3 the TiO2/fCNTs/GC modified electrode was modified with Horseradish Peroxidase and electrochemically characterized and applied to H2O2 detection with good results. But it was not used to an enzymatic substrate. Could the authors justify because this study was not necessary? Although the electrode modified with HRP is equivalent to the unmodified, it would be necessary to check whether it would be the same functioning by analyzing H2O2 generated in an enzymatic reaction.
On conclusion section, the authors could extend the principal analytical parameters of the H2O2 sensor (LOD, linear range, selectivity). Moreover, it should be highlighted the differences from another sensor.
Author Response
Author's Reply to the Review Report (Reviewer 3)
The manuscript describes a hydrogen peroxide (H2O2) sensor and biosensor based on modified multi-walled carbon nanotubes (CNTs) functionalizated with titanium dioxide (TiO2) nanostructures. Glassy carbon electrode was modified with TiO2-CNTs film and prussian blue as electrocatalyzer. The same sensor was also employed as the basis for H2O2 biosensors construction through further modification with Horseradish Peroxidase (HRP) immobilized at TiO2-CNTs film. Functionalized CNTs and modified TiO2-fCNTs were characterized by TEM, FTIR and XRD. Zeta potential was used to determine the surface charge on both CNTs samples. Solvent was optimized depending on the surface charge. PB-TiO2/fCNTs/GC modified was considered the best for the H2O2 detection in terms of operability. HRP-TiO2/fCNTs/GC modified GC electrodes showed electroanalytical response toward H2O2.
The manuscript is well organized, the introduction contextualizes the experimental section, the results section is concise, but the conclusion section could be extended by highlighting the advantages of this sensor and biosensor compared to existing (bio)sensors.
The authors appreciate your valuable suggestion; the conclusion has been improved in the new version of the manuscript.
I recommend this paper for publication in Nanomaterials. Nonetheless, some major changes must be covered:
On abstract, I think it would be appropriate to summarize the abstract.
The abstract has been improved in the new version of the manuscript.
On introduction section: The authors should extend the introduction including information and some references about composite electrodes based on graphite modified with glucose oxidase to glucose determination. This is a simple alternative to biosensors development. Some references are: Biosensors and Bioelectronics 78(2016)505–512; Microchemical Journal 119 (2015) 66-74.
The authors appreciate your valuable suggestion; the suggested references have been included in the new version of the manuscript.
On experimental section: Authors should complete the description of the materials and methods section. For example: purity of reagents, Sigma carbon nanotubes physical characteristics (as diameter and long) are necessary but they are not showed.
The requested information has been included in the new version of the manuscript (Section 2.1 and 2.2)
On line 109 (page 3). I think that the correct description for “pC-NTs dissolved” is “pC-NTs dispersed”.
The mistake was corrected
Please, specify the Nanocyl carbon nanotubes physical characteristics too.
The requested information has been included in the new version of the manuscript
On lines 178-183 (page 4). Authors should complete the description (model) of the equipment used.
The model of the equipment used has been included in the new version of the manuscript: Ch-instruments model 604 A Potentiostat
Information about reference electrode should be extended (model and internal solution concentration). The same for lines 190 of page 5.
Information about reference electrode has been included in the new version of the manuscript: Ag/AgCl (sat. KCl) reference electrode from Bioanalytical Systms(BAS)
On results section: The morphological and physical characterization of the different nanomaterials is complete and rigorous. Some figures could be removed anyway. For example, Figures 3, 4 and 5 could be placed in the supplementary section. The caption of Figures 1 and 2 should be completed. Images a and b are not detailed.
The figure was improved
On line 368-369 (page 12) the authors said that “The chronoamperometric detection H2O2 at PB-TiO2/fCNTs/GC electrode was performed at constant potentials of 0 V (Figure 9a) and +1 V (Figure 9b)”. Applied potential on oxidation process is very high and other species could be oxidized. Could the authors extended this part? Was an interference study conducted? How was the detection limit determinated?
The potentials to which the chronoamperometry experiments were performed were selected according to each voltammetric response (Figure 7), where it can be observed that there should be no interference in the response obtained
On section 3.3 the TiO2/fCNTs/GC modified electrode was modified with Horseradish Peroxidase and electrochemically characterized and applied to H2O2 detection with good results. But it was not used to an enzymatic substrate. Could the authors justify because this study was not necessary? Although the electrode modified with HRP is equivalent to the unmodified, it would be necessary to check whether it would be the same functioning by analyzing H2O2 generated in an enzymatic reaction.
We agree with your suggestion. Studies are currently being carried out in our laboratory to test the results obtained in real enzymatic substrates, which is part of future studies.
On conclusion section, the authors could extend the principal analytical parameters of the H2O2 sensor (LOD, linear range, selectivity). Moreover, it should be highlighted the differences from another sensor.
The authors appreciate your valuable suggestion. The principal analytical parameters of the H2O2 sensor have been considered in the Abstract and comparison of the characteristics of the modified electrodes with previous reports has been added in the new version of the manuscript (Section 3.4).

Round 2
Reviewer 2 Report
The authors have answered all the questions which were raised by the referee. Therefore, I recommend this version to be accepted for publication.
Reviewer 3 Report
The authors have addressed my requests and improved the quality of the manuscript. I approve the manuscript publication for the merit of its scientific content.